# Design Validation of a Low-Cost EMG Sensor Compared to a Commercial-Based System for Measuring Muscle Activity and Fatigue

**Anthony Bawa** 🆔 **and Konstantinos Banitsas** *🆔

Electronic and Electrical Engineering Department, Brunel University London, Kingston Lane, Uxbridge, London UB8 3PH, UK; anthony.bawa@brunel.ac.uk
* Correspondence: konstantinos.banitsas@brunel.ac.uk; Tel.: +44-(0)789-045-0501

**Abstract:** Electromyography (EMG) sensors have been used for measuring muscle signals and for diagnosing neuromuscular disease. Available commercial EMG sensor are expensive and not easily available for individuals. The aim of the study is to validate our designed low-cost sensor against a well-known commercial system for measuring muscle activity and fatigue assessment. The evaluation of the designed system was done through a series of dynamic exercises performed by volunteers. Our low-cost EMG sensor and the commercially available system were placed on the vastus lateralis muscle to concurrently record the signal in a maximum voluntary contraction (MVC). The signal analysis was done using two validation indicators: Spearman's correlation, and intra-class cross correlation on SPSS 26.0 version. For the muscle fatigue assessment, the root mean square (RMS), mean absolute value (MAV) and mean frequency (MNF) indicators were used. The results at the peak and mean level muscle contraction intensity were computed. The relative agreement for the two systems was excellent at peak level muscle contraction range (ICC 0.74–0.92), average 0.83 and mean level muscle contraction intensity range (ICC 0.65–0.85) with an average of 0.74. The Spearman's correlation average was 0.76 with the range of (0.71–0.85) at peak level contraction, whiles the mean level contraction average was 0.71 at a range of (0.62–0.81). In determining muscle fatigue, the RMS and MAV showed increasing values in the time domain, while the MEF decreased in the frequency domain. Overall, the results indicated a good to excellent agreement of the two systems and confirmed the reliability of our design. The low-cost sensor also proved to be suitable for muscle fatigue assessment. Our designed system can therefore be implemented for rehabilitation, sports science, and ergonomics.

**Keywords:** electromyography; low-cost sensor; muscle contraction; fatigue assessment

## 1. Introduction

An electromyography (EMG) sensor can provide information for muscular activity [1–3] and is used in clinical settings for diagnosing neuromuscular diseases. EMG sensors are used in research fields, such as ergonomics, sports science, and physiotherapy. The sensor measures the contraction of the muscle activity during physical exercise. The signals recorded from EMG sensors are non-periodic and continuous, which are generated by the skeletal muscles of the human body. The motor neurons cause the contraction of the muscle, which is responsible for movement. The muscle movement creates a depolarization, which is useful in assessing muscle stress and control. EMG sensors could be invasive or non-invasive, where invasive sensors are more accurate and with less noise compared to non-invasive sensors. Presently, there are some non-invasive commercial sensors in the market, which are expensive and may not be readily available for individuals. There has been some research in the design of a low-cost EMG sensor for clinical usage. Most of the designed low-cost EMG systems were stationary connected to computer ports. As mentioned above, earlier designed low-cost EMG sensors did not provide a standalone application that could

be used to monitor and record muscle signals. In the recent past, some researchers have developed low-cost EMG sensors which were applied in different fields. Beneteau et al. [4] designed an inexpensive wireless surface EMG sensor using a MSP430 microcontroller to record the lower arm muscle signal. The electromyogram features extracted from the designed system were used for pattern recognition. Wu et al. [5], designed a low-cost surface EMG sensor to recognize hand motion. The EMG system, miniaturized, had a wireless communication module to transmit the signal. Supuk et al. [6] presented a low-cost surface EMG sensor to record the muscle activities in a gait assessment. This method employed by the authors could be used to determine weak muscles. Surface EMG sensors are very useful for determining the strength of a muscle. Muscle fatigue can simply be described as the decline of muscle maximum force during contraction [7]. Muscle fatigue could occur in the cell of the muscle fiber or the nervous system. Currently, there are some methods which can be employed to detect muscle fatigue [8]. A surface electromyography sensor can be used to record the muscle function and electric muscle signal to detect muscle fatigue. This is performed by measuring the power or force measurement of muscle during maximal voluntary contractions (MVCs).

Some studies in the past have attempted to validate low-cost electromyography sensors with commercial-based systems. Heywood et al. [9] presented the validity of a fabricated low-cost EMG sensor on a microchip with a commercial system for dynamic contraction. The commercial device and the designed low-cost sensor were placed on the vastus lateralis muscle of volunteers performing different exercises. The signals were evaluated by means of a Teager–Kaiser energy operator (TKEO) and maximal voluntary contraction (MVC) of the muscle. The results indicated good agreement between the signal output of the two systems and the reliability of the low-cost sensor for measuring muscle activity. The inter-tester and intra-session were seen to be excellent with a peak contraction intensity (ICC > 0.99). Fuentes et al. [10] presented the validation of a designed low-cost EMG sensor with a commercial Delsys system. In the study, the authors used four validation indicators to estimate the similarity between the output signals of the two systems. Validation indicators used were Spearman's correlation, linear correlation coefficient (LCC), cross-correlation coefficient (CCC) and energy ratio of the signal. The Spearman's correlation had 0.60 as the average. The result showed an excellent agreement between the designed low-cost system and the commercial system. However, the limitation of this design system had to do with the noise of the hardware components and delay of the signal. Another study by Fuentes et al. [11], presented the validation of a low-cost surface EMG device for measuring muscle fatigue. In the study, 28 volunteers were prepared for a palpation test, where the designed low-cost sensor and commercial system were placed on the rectus femoris muscle. The mean absolute value (MAV), root mean square (RMS) and mean frequency (MNF) were used to evaluate muscle fatigue from the recorded signal. The results indicated that the low-cost EMG sensor can be used to determine muscle fatigue. Alejandro et al. [12] illustrated the validity of the mDurance system for measuring muscles activities by comparing it with a commercial sensor. In the study, volunteers were tested during isokinetic exercise at different speeds to determine the maximal voluntary contraction. The results indicated an excellent correlation of the vastus lateralis at (ICC > 0.81) and rectus femerois at (ICC > 0.76). This proved that mDurance is a valid tool for measuring muscle activity during dynamic contraction at different speeds. Jang et al. [13] demonstrated the reliability of a newly developed surface EMG sensor with a convectional EMG sensor during the voluntary isometric contraction exercise. The authors tested the newly designed surface electromyography device (PSL-EMG-Trl) with a convectional surface electromyography device (BTS-FREEMG1000). The signals obtained from the rectus femoris (RF) and biceps brachii (BB) muscles were compared using Pearson's correlation. The results indicated an excellent agreement with a high reliability for BB at range (ICC = 0.832–0.937) and RF at range (ICC = 0.814–0.957). This showed that the newly developed EMG device was effective for monitoring muscle activities during dynamic exercises.

Although some researchers have attempted to validate low-cost EMG sensors, the reliability of these sensors still have some challenges to be addressed. To the best of our knowledge, there is no study that has combined the validation of a low-cost sensor and its feasibility for fatigue muscle assessment. The novelty of this study is to test the reliability of our designed low-cost EMG sensor (MyoTracker) for measuring muscle activity and for determining muscle fatigue. This would be done by comparing it with a commercially available Delsys system.

## 2. Materials and Methods

The experimental procedure involved the capturing of muscle signal from the two devices concurrently. The first device was the newly designed low-cost EMG sensor (MyoTracker), and the second device was the commercial Trigno Avanti sensor from Delsys. The commercial system served as the gold-standard tool for measuring muscle signals.

### 2.1. Experiment Participants

In total, 7 participants were recruited for the study, and comprised 5 males and 2 females. The participants had no physical injury or any known deformity. The characteristics of the participants were as follows: age $25 \pm 7$ years, height $175 \pm 2.8$ cm, weight $72.4 \pm 5.6$ kg. The ethical approval was granted by the appropriate ethical committee. All the participants were given a consent form to sign prior to being recruited for the study. Participants were given an opportunity to withdraw from the study at any time without explanation.

### 2.2. Experiment Procedure

The experiment steps required to complete the exercises in Table 1 were explained to each of the participants. A demonstration was given first before they were prepared to conduct the exercises. Step 1: The risks involved in the study were explicitly explained to participants. A participant information sheet, which contains all information on the study, was given to the volunteers. Step 2: Participants filled in a questionnaire to collect information, such as their age, height, and weight. Step 3: Participants were prepared before placing the sensors on the vastus lateralis muscle. The hair on the skin was shaved and cleaned with muslin. This was done to minimize the interference from skin artifacts and remove dirt to give the sensors accurate readings. Our designed low-cost sensor and the commercial system were placed on the same region of the vastus lateralis muscle at 2 cm apart.

**Table 1.** Methodology for the exercises.

| Exercise | Instructions | Duration | Order |
|---|---|---|---|
| Frankenstein walk | Individual stands with legs together and one arm extended, steps to kick the opposite leg straight then stretches right arm. Then individual steps forward while repeating step. | 120 (s) | First exercise was the pre-trial, this was then followed by two trials recorded. |
| Band Sidewalk | An elastic band is place between the ankles of the subjects. Participants stand upright and bend a little downward at knee at 60-degree angle while holding their waist. Participants then perform a sidewalk on the left and right leg. | 120 (s) | First was the pre-trial stage, and then two-trials in a single lap with steps on each side of the legs in a sidewalk. |
| Wall sit | Individual participants lean against a wall with their feet planted firmly on the ground with feet 10 inches apart. The participants were asked to slide slowly down and then upward. | 120 (s) | First exercise was the pre-trial, and then two trials of step down and up. |
| Squat | Participants from a standing position lower their hips from the standing position and back to standing position at a comfortable speed. | 120 (s) | First pre-trial of squat and then two trials of squat at fast speed. |

### 2.3. Equipment for Muscle Signal Collection

The equipment for the study were the newly designed low-cost EMG sensor and the commercial based Trigno Avanti system from Delsys in Table 2. The low-cost sensor was designed using off the shelf hardware components, such as Arduino Uno, Myoware module,and HC-05 Bluetooth, to provide a wireless connection. An alkaline battery was used to provide power to the designed system. The Myoware sensor module was designed to output raw and envelop signal from the component manufacturers. The second equipment was the commercial non-invasive sensor from Delsys. The sensor electrodes were placed parallel to each other on a vastus lateralis muscle with a 2 cm distance. After the first trial, the sensors were interchanged for the second trial while still maintaining the 2 cm distance apart.

The vastus lateralis muscle shown in Figure 1 of the musculoskeletal system.

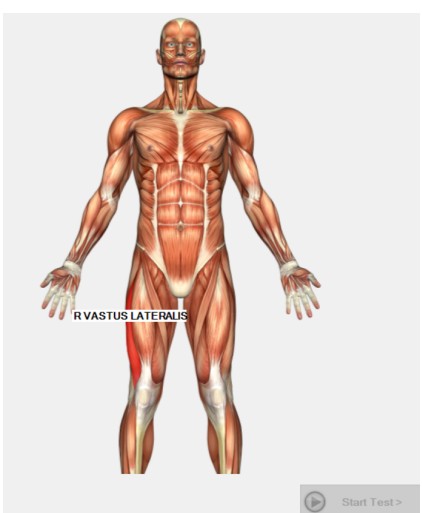

**Figure 1.** The vastus lateralis muscle.

Table 2 shows the technical features of both the designed low-cost EMG sensor and the Trigno Avanti EMG sensor [14].

**Table 2.** Technical specification of the two systems.

| Parameters | Low-Cost Sensor (MyoTracker) | Commercial Trigno Avanti Sensor |
| --- | --- | --- |
| Image | | |
| Price | $150 | $12,000 |
| Dimensions (mm) | MyoWare $52.9 \times 20.7 \times 5.1$ | $27 \times 37 \times 13$ |
| Weight (g) | Built EMG System 56.5 | 14.0 |
| Channels | 1 channel | $1\times$ EMG, up to $6\times$ IMU |
| Bandwidth (Hz) | 10–400 | 10–850 or 20–450 |
| Gain (V/V) | 201Rgain/1 kOhm | 300 |
| Sampling rate (Hz) | 333 | 1111 up to 2000 |
| Common Mode Rection Ratio (dB) | 110 | >80 |
| Operating Voltage (mV) | 3.3–5 | 11 |
| Contact electrode | Silver/Silver-chloride | 99.9 silver |
| Output mode | EMG Enveloped/Raw EMG | Raw EMG |

### 2.4. Data Processing and Analysis

EMGworks software [15] from Delsys was used for the signal processing. The software is a powerful tool for advanced research with a user-friendly interface for data analysis of EMG signals. After all the exercises were completed by the 7 participants, the signals from the two devices were saved on a csv file. This was exported to EMGworks for the signal processing. The analysis method adopted for this study was a combination of the methodology used by Heywood et al. [9] and Duffour et al. [16]. The steps are outlined as follows:

Step 1: The EMG signals obtained from the two systems were processed. This involved signal synchronization, filtering of the raw EMG signal and trimming. Filtering was done to denoise the raw signals from the two systems. The EMG signals were then rectified and enveloped.

Step 2: The computation of maximal voluntary contraction and normalization of the signals obtained from exercises conducted. MVC was calculated for every exercise performed by the participants in the study.

Step 3: The validation indicators were used for the signals obtained from the devices. The validation indicators compared the signals obtained for the two systems using SPSS version 26 package.

#### 2.4.1. Signal Filtering Process

We denoise the signals from the two devices separately by conducting different signal filtration for each device. The first signal filtering with the commercial system was done with the band pass Butterworth at 20–450 Hz, order 4. This is because a study by Wei et al. [17] illustrated that the EMG signal is known to be appropriate at 40–400 Hz. On the other hand, our designed low-cost EMG sensor was filtered using bandpass Butterworth at 45–55 Hz. The low-cost system can cater for the noise, which is centered around 50 Hz. A notch filter was included to remove the power line noise from the sensor.

#### 2.4.2. Signal Synchronization, Rectification and Trimming

The synchronization of the signals from the low-cost sensor and commercial sensor were overlapped for the exercises conducted by each participant. After the synchronization process, the signals were then trimmed to 44 s. The signals for the commercial device were then rectified using absolute mean values.

#### 2.4.3. Maximal Voluntary Contraction (MVC)

The maximal voluntary contraction occurred during the exercises performed by participants. The MVC values were computed for the two systems in the exercises conducted. The computation was done from the root mean square signal at the peak and mean levels for both systems.

The raw EMG signal is shown in Figure 2 and the filtered EMG signal is shown in Figure 3. The filtering of the noise was done using the band pass filter.

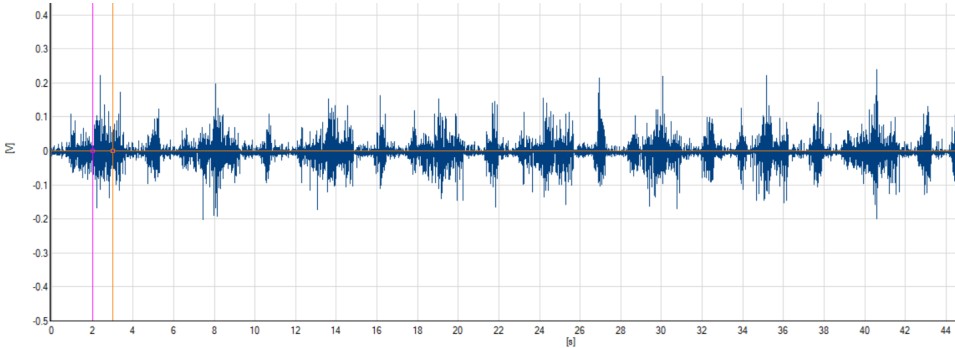

**Figure 2.** Raw EMG signal.

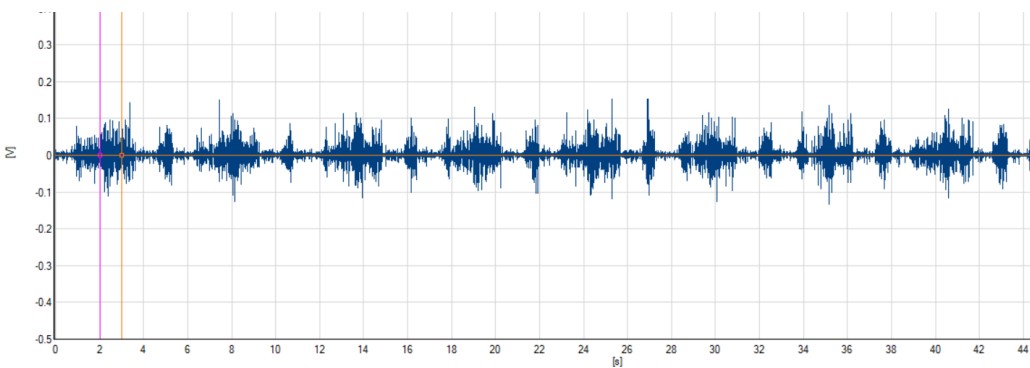

**Figure 3.** Filtered EMG signal.

The signal amplitude graph for the commercial system is shown in Figure 4, while the signal from the low-cost is shown in Figure 5

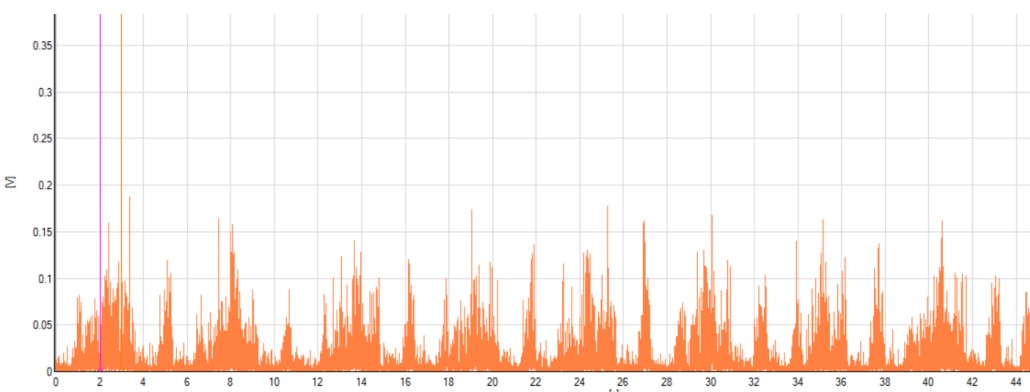

**Figure 4.** Commercial system.

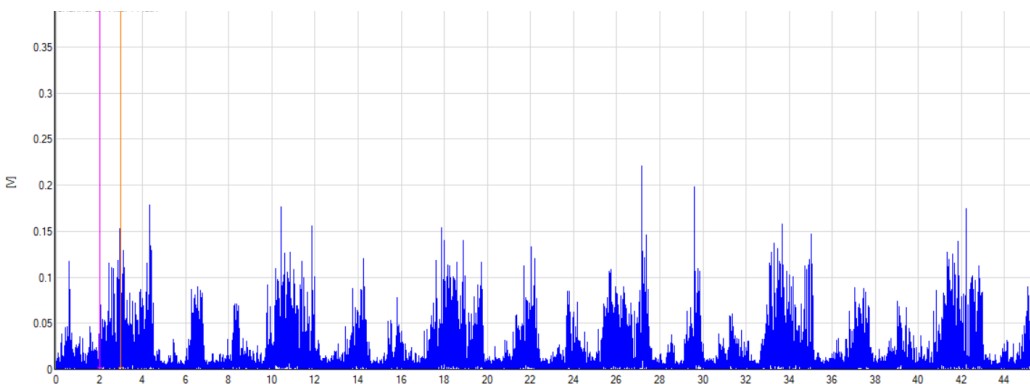

**Figure 5.** Low-cost system.

Figure 6 shows the synchronization of the signal between the two systems. Two validation indicators were used to achieve a successful evaluation of the designed low-cost sensor. The validators used were the Spearman's correlation and the intra-class correlation coefficient (ICC). The Spearman's correlation was used to measure the correlation between the signal of the low-cost sensor and the commercial system. The correlation can move between −1 and 1, where 1 indicates a positive correlation and −1 indicates a negative correlation. This can be mathematically expressed in Equation (1) as

$$\rho = 1 - \frac{6 \sum d_i^2}{n(n^2 - 1)} \tag{1}$$

where $d_i$ represents the difference between the two ranks in each observation, and $n$ is the number of observations.

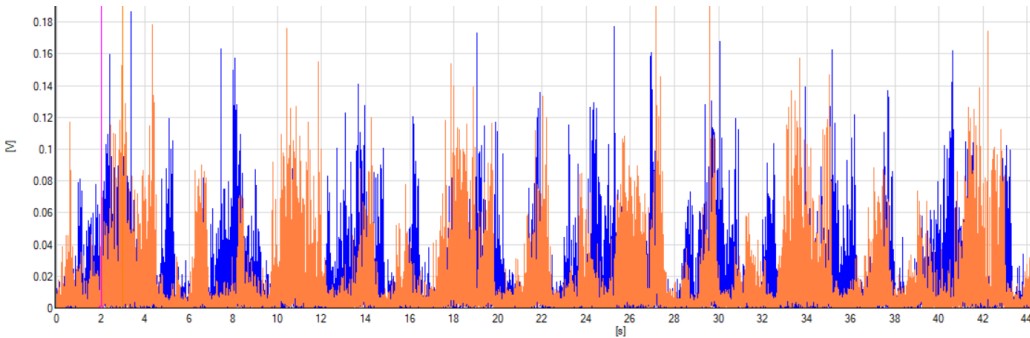

**Figure 6.** Synchronization of the two systems.

Intra-class correlation coefficient (ICC): The intra-class correlation coefficient was used to measure the relatedness and similarity between the signals of the two systems. This can be expressed in Equation (2) as,

$$ICC = \frac{S_b^2}{(S_b^2 + S_w^2)} \tag{2}$$

$S_b^2$ represents the cluster variance whiles $S_w^2$ represents within cluster variance.

### 2.5. Noise from Low-Cost System

Our designed low-cost EMG sensor was prone to some noise. The noise could affect the fidelity of the muscle signal capture. As the electrical signal travels through the tissues of the muscle fiber, there is some distortion of the signal. The ambient noise from the low-cost sensor is due to the electromagnetic radiation, which interferes with the quality of the signal captured. This is because the human skin surface is inundated with electric–magnetic radiation that is impossible to avoid. The noise was minimized by using a high pass filter to remove the distortion of the signal. The noise from the electronic components of the sensor can also be minimized by using silver chloride electrodes. This technique reduces the signal-to-noise ratio in the electric cables of the designed system.

The signal-to-noise ratio (SNR) for the low-cost system and the commercial system were computed. We found the SNR for the commercial system at peak contraction between 40–65 db, while the low-cost was between 10–32 db. The commercial system had a better SNR, compared to the low-cost sensor.

### 2.6. Electrode Skin Impedance

Electrode skin impedance is an essential parameter when recording EMG signals. This is because the electric impedance of the skin can interfere with the quality of the signal. The electrode skin impedance varies between the participants in the test. The room temperature for the exercise was 14 degrees Celsius with 76% humidity. This was to ensure that the participants did not sweat while conducting the exercises. The wet silver–chloride electrodes placed on the shaved and cleaned skin of the vastus lateralis muscle reduced the skin impedance. This enhanced the fidelity of the muscle signals captured.

### 2.7. Statistical Analysis

IBM SPSS statistics v.26 was used in the statistical analysis to compute the intra-class correlation coefficient and Spearman's correlation. The reliability for the two systems was calculated using a single measure of a two-way random model. According to Munro's descriptor [18], the reliability coefficient used in indexing the degree of reliability are (0.90–1.00) very high correlation, (0.70–0.89) high correlation, (0.50–0.69) moderate correlation, (0.26–0.49) low correlation, and (0.00–0.25) little or no correlation. The level of

agreement between the low-cost sensor and commercial system was examined for each exercise. Other parameters, such as the mean and standard deviation of signal output, for the two systems were also computed.

## 3. Results

In this study, seven participants completed four different exercises to test the validity of the newly designed low-cost EMG sensor. We tested for the reliability of our designed low-cost sensor for measuring muscle activity against the commercial Trigno Avanti sensor. The maximum voluntary contraction (MVC) values of the two systems were calculated for the exercises performed. The MVC was assessed at the peak and at the mean level for the designed low-cost sensor and commercial system. Table 3 shows the maximal voluntary contraction at peak level for both systems. The absolute agreement (ICC) was found between the range (0.74–0.92) with an average of 0.83 for the two systems. The minimum ICC was 0.74, and the maximum ICC was 0.92 at peak level. The Spearman's correlation coefficient at the peak level was in the range (0.71–0.85). The average was recorded to be 0.76 for the two systems. At peak level, the minimum value for the Spearman's correlation was 0.71 and the maximum was 0.85 for the two systems. For the mean level in Table 4, the absolute agreement range was (0.65–0.86). The average ICC was recorded to be 0.74 with the minimum at 0.65 and maximum at 0.86 for the two systems. The Spearman's correlation coefficient at mean level was between the range of (0.62–0.81) The average Spearman's correlation was calculated to be 0.71 with the minimum correlation at 0.62 and the maximum correlation recorded at 0.81. Other perimeters computed were the mean and standard deviation of the signal output for the two systems.

Concurrent validity: The absolute agreement (ICC), Spearman's correlation, the mean and standard deviation of the two systems at peak level shown in Table 3.

**Table 3.** Dynamic exercise at peak level.

| Exercise | Commercial Peak (MVC%) | Low-Cost Peak (MVC%) | Relative Agreement (ICC) | Spearman Correlation | Mean | SD |
|---|---|---|---|---|---|---|
| Frankenstein walk | 79 ± 28% | 68 ± 31% | 0.870 | 0.740 | 0.203 | 0.320 |
| Sidewalk | 82 ± 17% | 74 ± 20% | 0.740 | 0.720 | 0.519 | 0.314 |
| Wall Sit | 83 ± 20% | 80 ± 16% | 0.920 | 0.850 | 0.032 | 0.061 |
| Squats | 69 ± 31% | 58 ± 30% | 0.780 | 0.710 | 0.014 | 0.013 |

Concurrent validity: The absolute agreement (ICC), Spearman's correlation, mean and standard deviation for the commercial system and low-cost sensor at mean level shown in Table 4.

**Table 4.** Dynamic exercise at mean level.

| Exercise | Commercial Mean (MVC%) | Low-Cost Mean (MVC%) | Relative Agreement (ICC) | Spearman Correlation | Mean | SD |
|---|---|---|---|---|---|---|
| Frankenstein walk | 63 ± 37% | 57 ± 42% | 0.860 | 0.740 | 0.015 | 0.011 |
| Sidewalk | 74 ± 25% | 63 ± 36% | 0.670 | 0.620 | 0.334 | 0.336 |
| Wall Sit | 76 ± 24% | 70 ± 18% | 0.780 | 0.810 | 0.064 | 0.063 |
| Squats | 65 ± 18% | 52 ± 21% | 0.650 | 0.670 | 0.025 | 0.031 |

### 3.1. Signal Comparison for the Two Systems

The signals recorded from the two systems were processed and then overlapped together. The amplitude graphs for the low-cost and commercial system for the exercises conducted are displayed below.

The synchronization of the two systems shown in Figure 7 for Frankenstein walk and sidewalk shown in Figure 8.

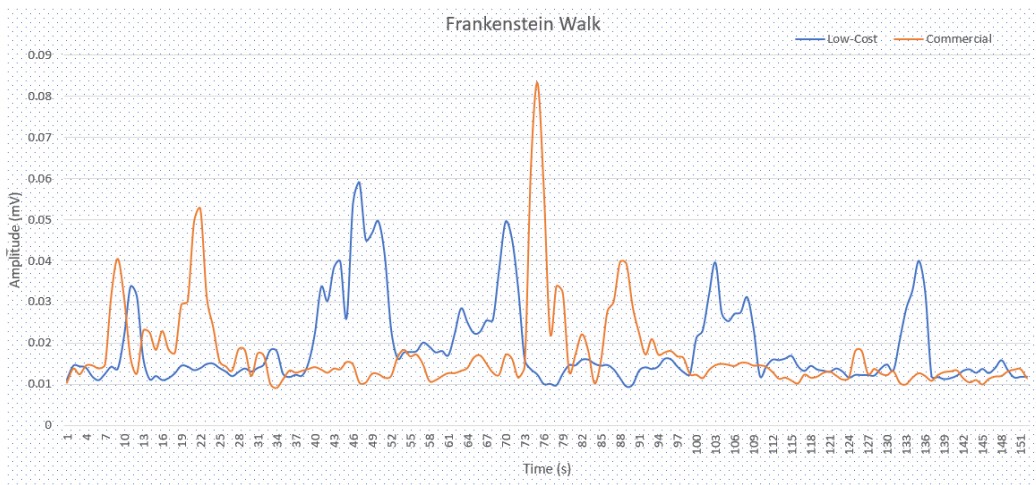

**Figure 7.** Amplitude graph of Frankenstein walk.

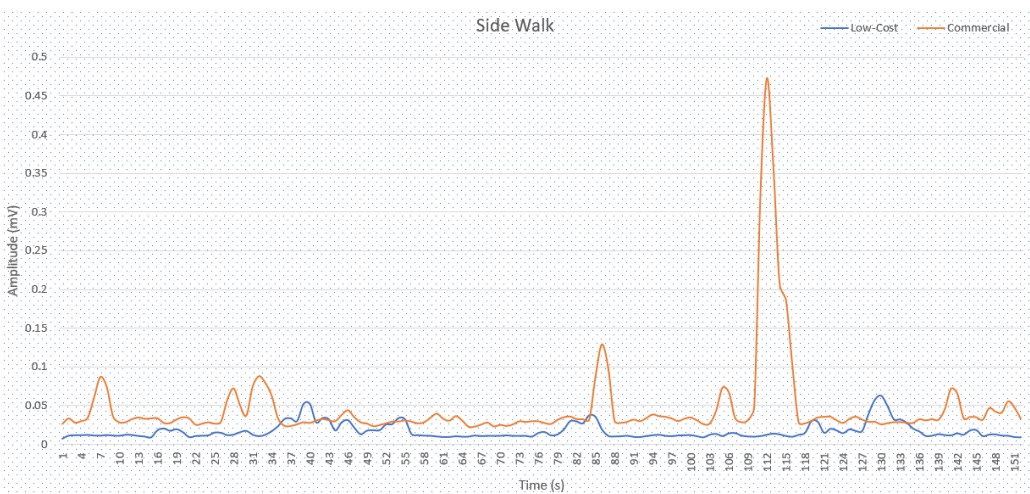

**Figure 8.** Amplitude graph of sidewalk.

Synchronization of the two systems in Figure 9 for wall sit and Figure 10 for squat.

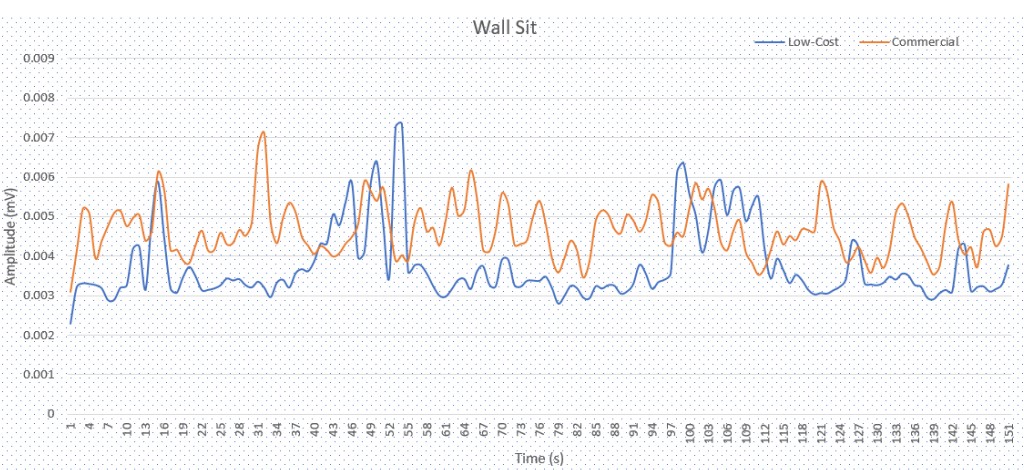

**Figure 9.** Amplitude graph of wall sit.

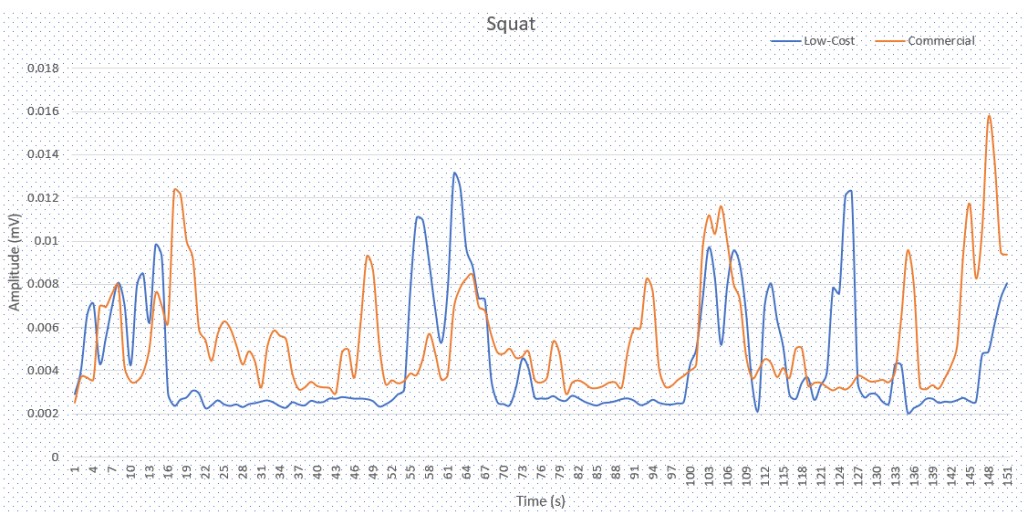

**Figure 10.** Amplitude graph of squat.

### 3.2. Muscle Fatigue Assessment

The muscle fatigue assessment can be determined by extracting signal features in the time domain and frequency domain. Fatigue detection in the time domain is estimated from features such as the root mean square (RMS) and the mean absolute value (MAV). Knowlont et al. [19] confirmed that an increase in the signal amplitude indicates the presence of muscle fatigue. Therefore, an increase in the signal of the root mean square and mean absolute values in the time domain indicate the presence of muscle fatigue, which can be detected by the low-cost EMG sensor. In the frequency domain, the mean frequency (MEF) can be combined with the RMS and MAV in the time domain to detect muscle fatigue. However, in contrast to the indicators used in the time domain, a decrease in the mean frequency will indicate the presence of muscle fatigue. For illustration, we selected the Frankenstein walk for the fatigue muscle assessment for the two systems. However, other exercises performed by participants could also be used for fatigue assessment. The three indicators used in determining muscle fatigue are expressed in the equations below:

$$RMS = \sqrt{\frac{1}{N} \sum_{i=1}^{N} x^2} \tag{3}$$

$$MAV = \frac{1}{N} \sum_{i=1}^{N} |x_i| \tag{4}$$

$$MNF = \frac{\sum_{j=1}^{M} f_j P_j}{\sum_{j=1}^{M} P_j} \tag{5}$$

The graphs below in Figures 11–13 illustrate the indicators used for the fatigue assessment. The root mean square, mean absolute value and mean frequency for the two systems are displayed in a boxplot.

Figures 11 and 12 illustrate an increase in the RMS and MAV values respectively in a box plot. Figure 13 also illustrates a decrease in the average frequency over time. All the above parameters indicate how the vastus lateralis muscle experiences fatigue for both systems.

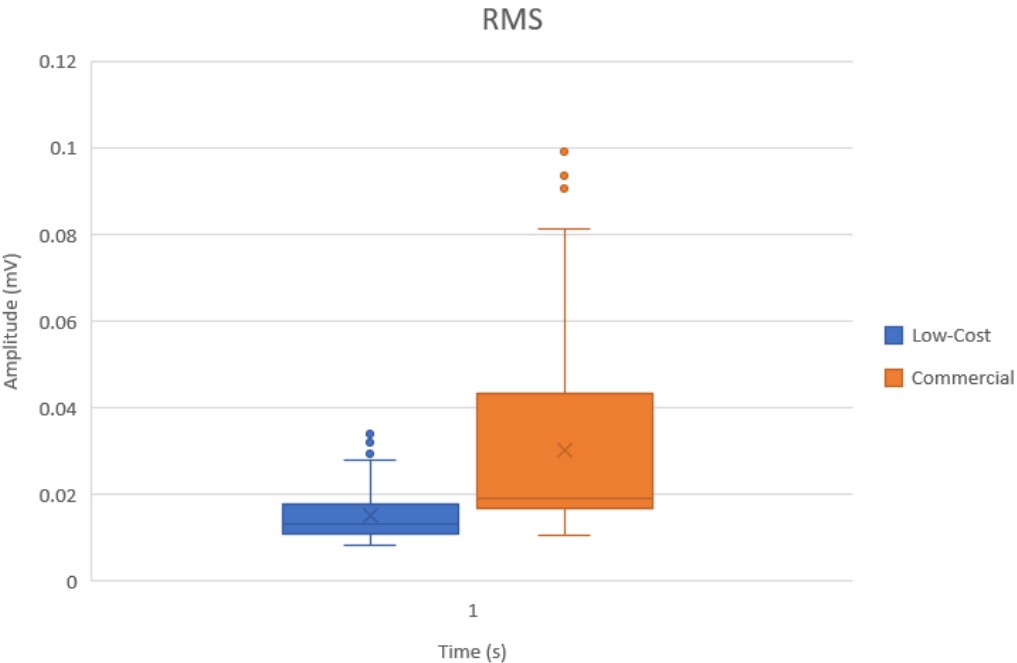

**Figure 11.** RMS of the systems.

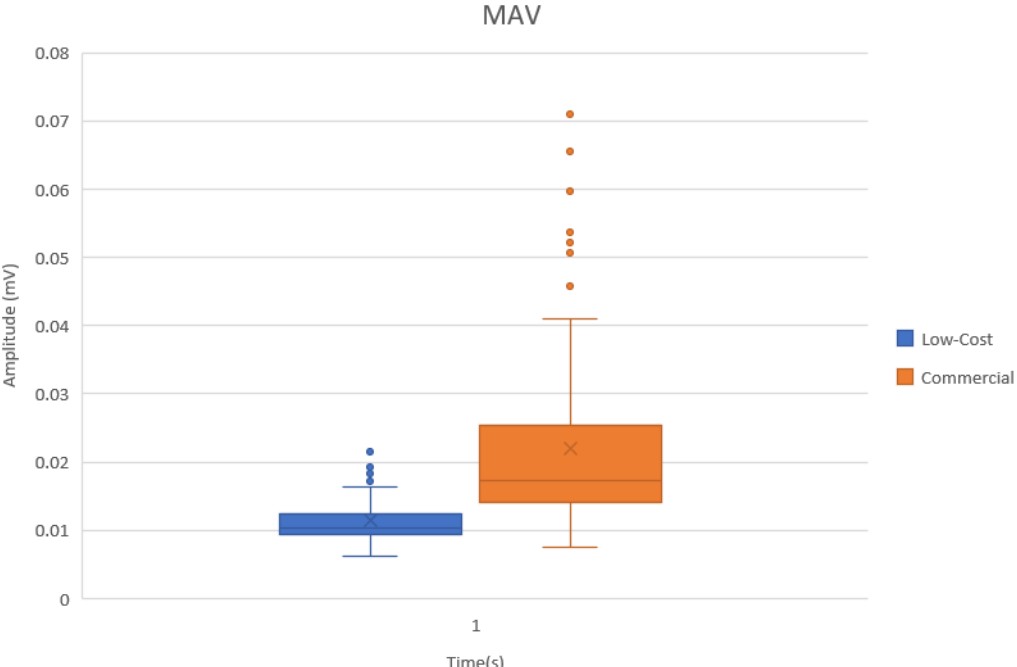

**Figure 12.** MAV of the systems.

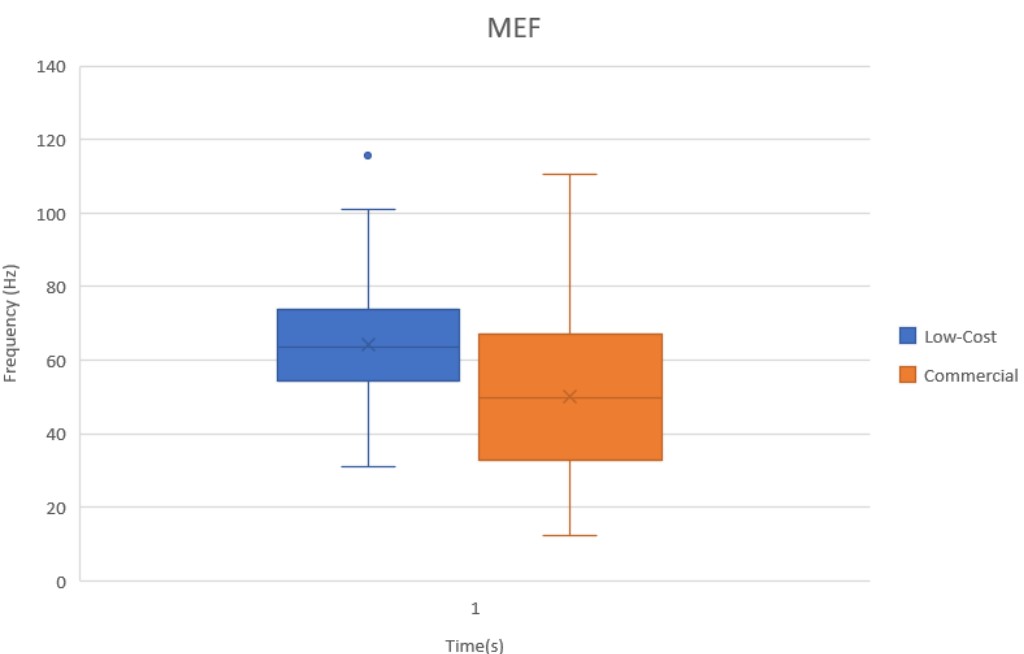

**Figure 13.** MEF of the systems.

## 4. Discussion

We tested the validity of our newly designed low-cost sensor for measuring muscle activity and determining muscle fatigue. To evaluate the reliability of the designed low-cost EMG sensor, we recruited seven participants to conduct dynamic exercises. The two systems were placed on the vastus lateralis muscle for each participant engaging in the exercises. It is possible to compare the muscle activity between one individual to another; however, caution must be taken when conducting such a comparison. The muscle contraction activity varies from one individual to another. The validation indicators used were the Spearman's rank correlation coefficient and the intra-class correlation (ICC). Spearman's correlation varies from 0 to 1, where 0 indicates that there is no similarity and 1 shows a very high similarity at 100% for the two systems. The results obtained indicated that there was a relative and absolute agreement at the peak level of ICC at a range of (0.74–0.92) for the two systems. The average intra-class correlation at peak level was recorded to be 0.83, which indicated a good to excellent agreement between the two systems. There is a high degree of reliability for the output signal of the low-cost sensor. Therefore, there is a significant correlation of our design EMG sensor and the commercial system. Additionally, at peak level, the Spearman's correlation was in the range of (0.71–0.85) with an average of 0.76, which indicates a very good correlation between the two systems. An average of 0.76 Spearman's correlation shows that there is a very good agreement of the two systems. At the mean level muscle intensity, the intra-class correlation was (0.65–0.86). The average correlation of the two systems at mean level was recorded to be 0.74, which indicates good agreement. The Spearman's correlation at the mean level was in the range of (0.62–0.81). The average correlation was found to be 0.71, which shows a good correlation for the two systems. The association between the two systems shows a good to excellent agreement of the systems. For the muscular fatigue assessment, we purposely selected the Frankenstein walk exercise with a series of indicators used to determine the presence of fatigue. The RMS and MAV shown in Figures 11 and 12, respectively, were used to determine the appearance of muscle fatigue. An increase in the RMS and MAV of the two systems indicates the moment that muscle fatigue begins to occur in the time domain. On the contrary, in the frequency domain, a decrease in the MEF shown in Figure 13 indicates the presence of muscle fatigue.

Comparing our design validation to previous work in [9,10], we developed an improved low-cost sensor which allows users to remotely connect their mobile devices to the

sensor. Users are able to view muscle signals on a mobile app in real time. The results from our validation had a higher correlation when compared to the work conducted by Jang et al. [13]. Again, our designed system employed some techniques to minimize the noise in the muscle signal. Moreover, the work performed in [11] did not consider noise mitigation of the design low-cost sensor. The techniques adopted in our study were able to minimize the noise in the low-cost sensor for a better signal-to-noise ratio in the validation process. The limitation of our design is the use of wet electrodes for the low-cost sensor. It is not always suitable to use wet electrodes in real-life applications. In future testing, we would include dry electrodes [20] and semi-dry electrodes [21] to improve the efficiency of our design. Conclusively, our developed system offers a reliable method for measuring muscle signal activity and is also suitable for determining muscle fatigue.

## 5. Conclusions

In this study, we tested the validity of our design low-cost EMG sensor (MyoTracker) against a commercial Delsys system. The main idea was to explore the viability of our low-cost design EMG sensor as an alternative to expensive commercial systems. Spearman's correlation and intra-class correlation were used as the validation indicators to test for the reliability of the designed system. The validation indicators showed good to excellent reliability of the low-cost EMG sensor for measuring muscle activity. In general, comparing our study to other previous works, we validated a newly portable low-cost EMG sensor for measuring muscle signal. The design system is also suitable for fatigue muscle assessment in a maximum voluntary contraction. With regards to the practical implication of the design system, we illustrated that our designed low-cost EMG sensor is quite promising for clinical implementation. In future works, our proposed design could be fabricated in a laboratory with further testing to ensure that it meets all the required specifications and standards of a commercial system. We estimated that our proposed design could be fabricated and commercialized for approximately $400. Factors that will account for the cost include profit margin, advertising, transportation and presentation. This could then be used for sport science, physiotherapy, and clinical rehabilitation.

**Author Contributions:** A.B. and K.B. contributed proportionally to this study. A.B. did the main write up, whiles K.B. did the editing, structuring of the paper and conclusion. All authors have read and agreed to the published version of the manuscript.

**Funding:** This research received no external funding.

**Institutional Review Board Statement:** The study was conducted in accordance with the Declaration of Helsinki, and approved by Ethics Committee of Brunel University London, reference code 35807-MHR-Apr/2022-39291-3 on the 10/06/2022.

**Informed Consent Statement:** Informed consent was obtained from all participants involved in the study.

**Data Availability Statement:** Data available on request to corresponding author due to ethical restrictions.

**Acknowledgments:** Authors would like to thank all volunteers for their contribution to the study.

**Conflicts of Interest:** The authors declare no conflict of interest.

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
