# Peer review of "Design Validation of a Low-Cost EMG Sensor Compared to a Commercial-Based System for Measuring Muscle Activity and Fatigue"

_sensors, doi:10.3390/s22155799_

Round 1

Reviewer 1 Report

The introduction is correct and comprehensive, perfectly situating the conceptual framework of the study to be carried out.
The methodology is appropriate for the study being carried out, as is the presentation of the results.
However, the discussion only contemplates the analysis of the results of the study carried out, and it would have been interesting to compare this analysis with that offered by other studies carried out with the equipment presented in the introduction.
This reiteration is also found in the conclusions.The discussion and conclusions should be rewritten because the study is correct and interesting.

Reviewer 2 Report

The paper presents a low-cost sensor validation with respect to a commercial one.

There are some aspects that need to be taken care of before further processing:

There are some typos in the text that should be corrected

-line 138 fellows instead of follows, table 1

-table 1” a band is place in the ankle of the two legs” should be: an elastic band is placed between the ankles of the subject…

Figure 1 should be referred if its not a original wok, and vastus lateralis should be indicated using an arrow.

The quality of the Figures 2,3,4,5 should be enhanced because are not readable.

All figures should contain measuring units (2, 3, 4, 5).

Terms of eq 1 and 2 are not explained in the text of the paper.

Tables 3 and 4 contain values that have variations up to 42 % are these values adequate, please make a comparison between data obtained by other researchers to fully assess the data obtained.

The center of the paper is a low-cost sensor with an estimated value of 150 USD, compared with a 12000 USD one. How much would you consider the price of your sensor would be in order to become a commercial product?

In order to obtain a product usable in patient evaluation the sensor should be tested in relevant conditions. Please mention future work required to achieve this level of technology readiness.

Best regards.

Reviewer 3 Report

The authors presented design validation of a low-cost EMG sensor compared to a commercial based system for measuring muscle activity and fatigue. These findings are very important for emerging EMG applications. I have identified the main points for consideration below:

1.     This manuscript has some spelling typos, style errors and grammatical errors. Pleases correct them in the revised manuscript.

2.     The authors should compare the signal-to-noise ratio (SNR) of EMG signals recorded the low-cost EMG sensor and the commercial one.

3.     Electrode-skin impedance is an important parameter for recording EMG signals. So, the electrode-skin impedance is recommended to be added in the revised manuscript.

4.     The cost of the proposed sensor should be given in the manuscript.

5.     The novelty of this study should be clarified in the manuscript.

6.     In this work, EMG signals were recorded by wet electrodes, which is not very convenient for real-life applications. In the discussion section, I suggest the authors discuss the new electrode solutions for EMG recording, including semi-dry electrodes or dry electrodes. Some recent related references are recommended to cited, such as J. Neural Eng. 17 (2020) 051004; J. Neural Eng. 13 (2016) 046021.

Round 2

Reviewer 2 Report

Dear authors, 

all my comments have been answered, I have no further comments.

Best regards.

Reviewer 3 Report

The authors have substantially improve the manucript. Please replace the Ref 22 with a recent related reference  J. Neural Eng. 18 (2021) 046016.